# ASA Allergy and Desensitization Protocols in the Management of CAD: A Review of Literature

**DOI:** 10.3390/jcm12175627

**Published:** 2023-08-29

**Authors:** Monica Verdoia, Rocco Gioscia, Matteo Nardin, Giuseppe De Luca

**Affiliations:** 1Division of Cardiology, Nuovo Ospedale degli Infermi, ASL Biella, 13900 Biella, Italy; gioscia.r@gmail.com; 2Division of Internal Medicine, Spedali Civili, 13900 Brescia, Italy; mecionardino@hotmail.com; 3Divisione di Cardiologia, AOU “Policlinico G. Martino”, 98122 Messina, Italy; giuseppe.deluca@unime.it; 4Dipartimento di Medicina Clinica e Sperimentale, Università di Messina, 98122 Messina, Italy; 5Division of Cardiology, IRCCS Hospital Galeazzi-Sant’Ambrogio, 20151 Milan, Italy

**Keywords:** aspirin, allergy, desensitization, coronary artery disease

## Abstract

Acetylsalicylic acid (ASA) hypersensitivity still represents one of the major deals for patients with atherosclerotic cardiovascular disease (ASHD), especially for those requiring percutaneous coronary interventions in the absence of validated alternative options. Despite symptoms after ASA administration being reported in 6–20% of cases, true ASA allergy only represents a minority of the patients, pointing to the importance of challenge tests and potential strategies for tolerance induction. ASA desensitization protocols were proposed several decades ago, with accumulating the literature on their use in patients undergoing PCI either for chronic disease or acute coronary syndromes. Nevertheless, the promising results of the studies and meta-analyses have not been validated so far by the support of large-scale randomized trials or unique indications from guidelines. Therefore, ASA desensitization is still largely unapplied, leaving the management of ASA hypersensitivity to the individualized approach of cardiologists.

## 1. Introduction

Acetylsalicylic acid (ASA) represents the pillar therapy in patients with ascertained atherosclerotic cardiovascular disease (ASHD) to manage both the acute phase and to prevent recurrent events [1,2,3].

In fact, ASA benefits in primary prevention have not been able to overweight the increased risk of bleeding complications, leading most scientific societies and practical guidelines to recommend against its initiation in the absence of ASHD [4]. 

On the contrary, in patients with established atherosclerotic disease involving either the cerebral, coronary, or peripheral district, ASA is recommended as a first-line strategy on top of lifestyle measures in guidelines worldwide, to be initiated as soon as possible in patients with acute ischemic events and to be continued lifetime [5]. 

In particular, the role of ASA is even more relevant in patients with acute coronary syndromes undergoing percutaneous coronary interventions (PCI), where a dual antiplatelet therapy (DAPT) is required, since the only guidelines-approved DAPT regimen, so far, is represented by the combination of ASA with a P2Y12 inhibitor [6].

However, while several molecules have been developed within most of the pharmacological families of cardiovascular drugs, including statins, beta-blockers, and other antiplatelet agents, allowing the clinicians the choice according to different properties and patients’ tolerance, ASA still represents a unique as an antithrombotic drug. In fact, none of the other available non-steroidal anti-inflammatory drugs (NSAIDs) have displayed the same capability of selective and irreversible blockage of the platelet cyclooxygenase 1 (COX-1), thus offering the same antithrombotic effects [7]. 

Nevertheless, ASA also represents one of the drugs most commonly associated with discontinuation for reduced tolerance. The National Institute for Clinical Excellence (NICE) in the U.K. has defined aspirin intolerance as either a proven hypersensitivity to aspirin or a history of severe indigestion caused by low-dose aspirin. The prevalence of aspirin intolerance is between 6% and 20%, with ‘true’ aspirin hypersensitivity occurring in 0.6–2.4% of the general population [8].

Allergy to aspirin is reported in 1.5–2.6% of patients presenting with CHD, although not representing, in the majority of cases, a true immune-mediated allergy, but rather a kind of intolerance, a direct consequence of the mechanisms of action of ASA [9]. 

Therefore, discriminating between the different forms of ASA hypersensitivity and the identification of strategies for the management of these patients represent a challenge for modern cardiology that has not been overcome by recent developments. 

## 2. Classification of ASA Hypersensitivity

ASA sensitivity can include a true immune-mediated allergy or a variety of symptoms depending on the pharmacological effect of ASA itself. In fact, the primary mechanism is believed to be inhibition of the cyclooxygenase 1 (COX-1) enzyme, inducing a shift in the cytokines pathways leading to an overproduction of leukotrienes and a reduction in prostaglandins, thus producing an imbalance leading to symptoms (Figure 1) [10]. Therefore, patients with aspirin sensitivity often display cross-reactions with other nonselective NSAIDs that inhibit the COX-1 enzyme.

Non-allergic aspirin sensitivity is divided into two main subgroups: the bronchospastic and the urticaria/angioedema type [11]. The district involved is generally conditioned by a pre-existing condition of the patient, already displaying a tendency toward developing cutaneous or respiratory disorders, which are only exacerbated by the administration of NSAIDs. 

### 2.1. Cutaneous Urticaria and Angioedema

Patients with chronic spontaneous urticaria can experience cutaneous reactions within hours of NSAID ingestion, being reported in 12–30% of the patients, which is defined as NSAID-exacerbated cutaneous disease (NECD). The shift in cytokine pathways induced by ASA, with reduced prostaglandin E2 levels, promotes mast cell activation and histamine release. 

However, similar to NECD, healthy individuals without a history of chronic urticaria can develop the same cutaneous symptoms (pruritus, urticaria, angioedema) following exposure to COX-1-inhibiting NSAIDs, leading in more severe cases to hypotension, which can mimic anaphylaxis. 

### 2.2. Respiratory Manifestations 

NSAID-exacerbated respiratory disease (NERD), formerly named aspirin-exacerbated respiratory disease (AERD), manifests primarily as bronchial obstruction, dyspnea, and nasal congestion or rhinorrhea in patients with a pre-existing history of respiratory disease. 

The frequency of aspirin intolerance has been described as 6.18% in patients with perennial rhinitis and 14.68% in patients with nasal polyps, the latter representing one of the populations at higher risk of intolerance [12]. The key pathogenic event for aspirin sensitivity is the change in the leukotriene pathway for arachidonic acid metabolism releasing high amounts of leukotrienes LTC4, LTD4, and LTE4. Cysteinyl leukotrienes (CysLT) promote bronchoconstriction, increase vascular permeability, mucous hypersecretion, and eosinophil chemotaxis [13]. 

### 2.3. Immune-Mediated ASA Allergy 

The proper forms of ASA allergy, derived from the direct reaction of the immune system to the molecule, can also be divided into two different forms: -The single-NSAID-induced urticaria/angioedema or anaphylaxis (SNIUAA) represents an acute (from minutes to hours from the administration) IgE-mediated allergic response, displaying all the traditional features of an allergic reaction.-The single-NSAID-induced delayed hypersensitivity reactions (SNIDHR), instead, occur more than 24 h after drug exposure and are secondary to drug-specific T lymphocyte effects. Clinical manifestations can range from a simple rash to severe conditions, including Stevens–Johnson/toxic epidermal necrolysis, nephritis, pneumonitis, and aseptic meningitis. In these patients, therefore, re-exposure to NSAIDs is contra-indicated [14].

## 3. Pathophysiological Principles for ASA Desensitization

The pivotal role of ASA in ASHD, and especially in patients with coronary artery disease, has driven the interest of cardiologists toward the identification of strategies for the management of patients with ASA intolerance, exploring either its substitution or the possibility of sensitivity overcoming. 

As said, no drug has been able, so far, to display the same antithrombotic effects of ASA. In the recently released Guidelines of the European Society of Cardiology on Chronic Coronary Syndromes (CCS), Knuuti et al. [15] indicated to use clopidogrel instead of aspirin in patients with chronic coronary syndrome (Class I level of evidence B) but prasugrel or ticagrelor monotherapy for a DAPT after percutaneous coronary intervention (PCI) (Class IIb recommendation, level of evidence C). 

In these patients, however, testing for true allergy, including the administration of an ASA challenge dose, could offer the opportunity to reduce the number of patients where ASA is really contra-indicated. 

Another option for overcoming ASA intolerance, however, is provided by desensitization protocols, representing a valid and easy-to-obtain alternative, especially for those patients not displaying a true immune-mediated allergy. 

ASA desensitization was first described in 1922 in the setting of respiratory disease, being performed in a 37-year-old woman with asthma and nasal polyps who noted worsening asthma symptoms following the ingestion of aspirin [16], being described as the administration of infinitesimal doses of ASA, progressively increasing in a continuous fashion, allowing then the patient to tolerate subsequent doses of aspirin without complications. 

The cellular and molecular mechanisms driving the improvement following aspirin desensitization/maintenance therapy remain largely speculative. In fact, despite the fact that it might be speculated that the protocol could induce a modulation and re-balancing in the cytokines production, several studies have shown no reduction in the concentration of leukotrienes in patients undergoing desensitization and tolerating ASA [17,18]. On the contrary, desensitization has been shown to reduce the expression of the receptor for CysLT and its sensitivity [19], or, in addition, ASA could also directly act on inflammatory cells, preventing their recruitment. 

So far, several desensitization protocols have been proposed, with different administration routes, dose escalation times, and starting doses.

In patients with coronary artery disease, several studies and meta-analyses have shown that these procedures are safe and effective in over 90% of patients [20,21,22], both in chronic and acute coronary syndrome patients, although standardization and evidence from large-scale randomized trials are still lacking. 

## 4. Overview of Available Desensitization Protocols

ASA desensitization protocols comprise oral and intravenous protocols with starting doses ranging from 0.1 to 10 mg and different therapeutic steps, resulting in protocol duration from 2 to over 5 h. 

The features of different proposed protocols are displayed in Table 1 [23,24,25,26,27,28,29,30,31,32,33,34,35,36,37,38,39,40,41,42].

As shown, so far, only one intravenous protocol has been published, offering the advantages of allowing its performance even in urgent settings, including intubated patients or those with ST-segment elevation myocardial infarction, in whom drug absorption is impaired, and fast onset of action is required. 

Oral protocols for ASA desensitization include a wide variety of schemes, with either < or >6 ASA doses. Rossini’s protocol has the greatest sample size and the best efficacy and safety data; however, subsequent cohorts and meta-analyses were published for a total population of over 1000 desensitized patients. All the protocols were safe and effective in the achievement of ASA tolerance, without relevant differences between intravenous (98%: 97.9–98%) and oral or less versus more fractionated (95.8% [95.4–96.3%] versus 95.9% [95.2–96.5%]) strategies [20]. 

Moreover, while the majority of initial protocols included only stable patients, more recent data have also been achieved among patients presenting with acute coronary syndrome, showing similar efficacy, >96% in a total of five studies (n = 330 patients) [25,27,43]. 

In a previous meta-analysis by Verdoia et al. [21], hypersensitivity symptoms occurred in 9.7 [6.1–13.3]% of the patients. All adverse reactions were safely managed through corticosteroids, antihistamines, bronchodilators, and/or adrenaline, according to the clinical setting. In 11 of these patients (38.9%), slowing the protocol or restarting another ASA challenge could successfully achieve the tolerance, allowing to maintain ASA therapy. 

The most recent largest-scale meta-analysis, by Chopra et al. [22], also confirmed no statistical difference in outcomes between protocols ≤2 h and >2 h in duration (96.3 [92.3 to 100.3]% vs. 97.2 [94.6 to 99.8]%; *p* = 0.71). Protocols with > 6 dose escalations were associated with higher success rates compared to those with ≤ 6 doses (99.2 [97.9 to 100.4]% vs. 95.4 [93 to 97.8]%; *p* = 0.007), confirming the potential advantages of a slower escalation of ASA therapy. At a follow-up, mostly over 12 months, no hypersensitivity events or ASA discontinuations were reported. 

## 5. Management of ASA Allergy in the Literature and Guidelines

Despite the clinical relevance of ASA allergy in ASHD, and especially among patients undergoing PCI, where DAPT is indicated, ASA hypersensitivity is still largely neglected in the literature and guidelines, not providing unique indications for their management and leaving alternative strategies to the individual discretion. However, most cardiologists are still largely unaware of the desensitization options, preferring to abstain from ASA administration and, thus, often addressing these patients with surgical rather than percutaneous revascularization. In fact, several surveys have been published to explore the use of ASA desensitization. In a large study among 116 PCI centers in the UK [44], 87.9% of them reported the lack of dedicated protocols, while only 12.5% routinely performed desensitization. In a more recent study, Bianco et al. [20,45] collected the answers from 86 physicians, of whom 56% managed aspirin hypersensitivity changing the therapeutic regimen (e.g., clopidogrel monotherapy and indobufen), while only 42% performed desensitization protocols. 

Indeed, despite the promising results of ASA desensitization in terms of safety and effectiveness and its wide applicability, with easy nurse-based management, no mention of desensitization protocols is made in current guidelines about myocardial revascularization [15], preferring instead to address single P2Y12 inhibition. However, such an option is certainly less supported by the literature data being explored so far, only in one study on 70 patients, mainly with acute coronary syndrome. The authors tested the safety and efficacy of prasugrel and ticagrelor monotherapy in patients with aspirin intolerance, showing at one-year follow-up a new clinically relevant adverse event in 25.7% of the patients [46]. 

Nevertheless, given the large interest and clinical relevance of the topic, and the limited data with alternative strategies, ASA desensitization appears, at present, the strategy with a more solid and larger scale background of data. Therefore, despite not being supported by guidelines’ indications, the application of desensitization protocols should be advocated in the majority of CAD patients reporting cutaneous respiratory ASA allergy, without anaphylaxis, in order to enhance the possibility of ASA prescription in these patients. 

Thus, greater attention should be played to spreading the knowledge of these potential options for the management of ASA allergy in patients undergoing PCI, although dedicated randomized trials, which are not planned so far, are certainly deserved to confirm initial findings and for the definition of the best-performing strategy, among the proposed protocols. 

## Figures and Tables

**Figure 1 jcm-12-05627-f001:**
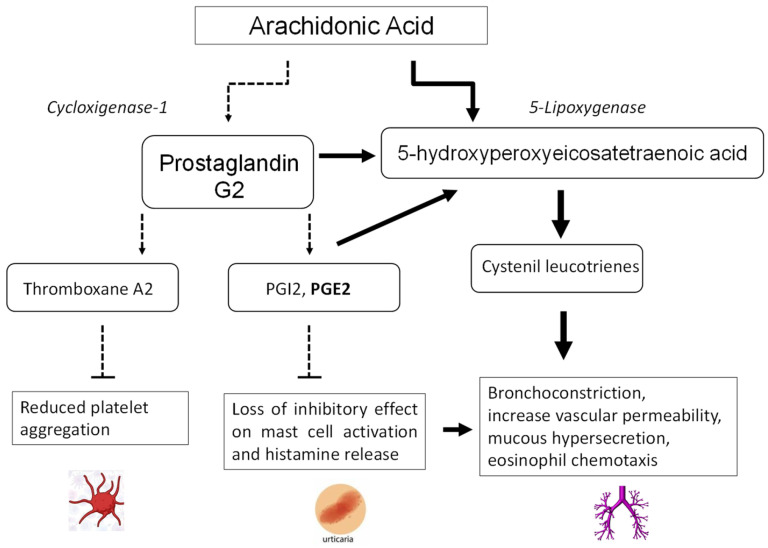
Mechanisms of aspirin intolerance.

**Table 1 jcm-12-05627-t001:** Characteristics of included studies.

Study	Patients n	Indication to ASA	Protocol Type	Protocol Duration	ASA Cumulative Dose	Protocol Description (Dosage, mg)	Premedication
Christou et al. [26]	11	PCI for stable CAD/ACS	Oral	3.5	648.4 mg	0.1, 0.3, 10, 30, 40, 81, 162, 325	-
Cortellini et al. High Risk [27]	31	Planned PCI	Oral	3.5	150 mg	0.1, 1, 1.5, 2, 3, 4, 5, 10, 15, 25, 35, 50	-
Cortellini et al. Low Risk [27]	30	Planned PCI	Oral	3	160 mg	10, 15, 25, 20, 50	-
Dalmau et al. [28]	5	PCI for ACS	Oral	2.5	189.4 mg	0.1, 0.2, 1, 3, 10, 25, 50, 100	-
De Luca et al. [25]	43	PCI for stable CAD/ACS	Endovenous	4.5	500 mg	1, 2, 4, 8, 16, 32, 64, 128, 250	-
Diez et al. [29]	13	PCI for stable CAD/ACS	Oral	2.5	189.4 mg	0.1, 0.2, 1, 3, 10, 25, 50, 100	Antileukotrienes (24 h before and 1 h before) and dexchlorpheniramine (1 h before) in the patient with a history of prior anaphylaxis
Hobbs et al. [30]	13	PCI	Oral	3.5	799 mg	1, 2, 4, 8, 15, 30, 50, 81, 121, 162, 325	Prednisone, montelukast, and cetirizine from 12 h prior to protocol
Lee et al. [31]	24	ACS or PCI for stable CAD/ACS	Oral	3	155 mg	5, 10, 20, 40, 80	-
Mc Mullan et al. [32]	23	Coronary artery disease (CAD) or a cardiac procedure	Oral	2	636 mg	1, 10, 20, 40, 80, 160, 325	-
Ortega Loayza et al. [33]	3	PCI	Oral	4	227.5 mg	0.5, 1, 2, 4, 8, 16, 32, 64, 100	Diphenhydramine, 50 mg
Rossini et al. [23]	26	Admitted for cardiac catheterization	Oral	5.5	176 mg	1, 5, 10, 20, 40, 100	-
Silberman et al. [25]	16	Recent percutaneous coronary intervention	Oral	2.5	160 mg	5, 10, 20, 40, 75	-
van Nguyen et al. [34]	3	PCI for stable CAD/ACS	Oral	6–10 days	-	0.001/10 to 100 mg	Fexofenadine
Veas et al. [35]	4	ACS	Oral	5	176 mg	1, 5, 10, 20, 40, 100	-
Wong et al. [24]	11	CAD (1 or pulmonary embolism)	Oral	3	652.4 mg	0.1, 0.3, 1, 3, 10, 30, 40, 81, 162, 243, or 325	Loratidine, cetirizine, hydroxyzine, or diphenhydramine
Vlachos et al. [36]	48	ACS	Oral	4	500 mg	0.1; 12.5 mg; 25 mg; 50 mg; 100 mg; 250 mg; 500 mg	None (prick test before)
Vega et al. [37]	11	ACS or stable CAD	Oral	2.5	189.4 mg	0.1, 0.2, 1, 3, 10, 25, 50, 100	-
Jackson [38]	24	ACS or stable CAD	Oral	2	160 mg	5, 10, 20, 40, 75	-
Cordoba-Soriano et al. [39]	24	ACS	Oral	2.5	189.4 mg	0.1, 0.2, 1, 3, 10, 25, 50, 100	Prednisone + cetirizine
Rossini el al. ADAPTED registry [40]	330	ACS or stable CAD	Oral	5.5	176 mg	1, 5, 10, 20, 40, and 100 mg	-
Cortellini et al. [41]	310	ACS or stable CAD	Oral	5	100.1	0.1, 1, 2, 3, 4, 5, 10, 15, 25, 35	-
Al-Ahmad et al. [42]	23	ACS	Oral	(A) 3.0 (B) 1.0 (C) 2.25 (D) 2.25 (E) 2.0 (F) 0.5 (G) 2.5 (H) 1.5 (I) 2.0	(A) 207 (B) 83 (C) 81 (D) 84 (E) 82 (F) 82 (G) 246 (H) 81 (I) 84	(A) 21, 21, 21, 21, 41, 41, 41 (B) 21, 21, 41 (C) 10, 21, 21, 29 (D) 21, 21, 21, 21 (E) 41, 41 (F) 41, 41 (G) 41, 41, 41, 41, 41, 41 (H) 10, 21, 21, 29 (I) 21, 21, 21, 21	Antihistamine

## Data Availability

Not applicable.

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
