# Peer review of "ASA Allergy and Desensitization Protocols in the Management of CAD: A Review of Literature"

_jcm, 2023, doi:10.3390/jcm12175627_

Round 1

Reviewer 1 Report

The paper is well written and deal with An interesting topic. 

I have only one minor revision: 

Page 5 raws 178-179 Cite also Eur Heart J 2020 doi: 10.1093/eurheartj/ehz889

Thank you

Author Response

The reviewer’s comments have been greatly appreciated and fully considered in the revision of the manuscript

The paper is well written and deal with An interesting topic.

I have only one minor revision:

Page 5 raws 178-179 Cite also Eur Heart J 2020 doi: 10.1093/eurheartj/ehz889

REPLY 1: Thank you, we have updated our references as suggested, (added REF.45)

Reviewer 2 Report

The submitted review with title: "ASA allergy and desensitization protocols in the management of CAD" describes an important clinical point for our daily life in how to deal with this phenomenon. As a pathology it has been underrated in the last years. However, its relevance is being recently more noted. 

The management of CAD and Aspirin intolerance is challenging, as a proper practical guideline based approach on how to manage it, in this field is still vague. 

The present paragraphs are clearly presented and logically sorted out. 
The authors managed to properly analyse the current literature in this field and bring out this thoroughly thoughted review. The syntax of the manuscript is correctly done.

No major are to be stressed out. However two important points should be addressed to: 

a) Due to the vast majority of reported adverse reactions to aspirin weighed with the overall benefit of aspirin in CAD patients, we believe aspirin desensitization protocols serve as an excellent method of allowing CAD patients to be more tolerable to aspirin. Therefore by taking the nature of being a review in this fied --> What is your practical appraoch in managing this sensitive matter ? What could be to fullfill the review, a practical management approach to CAD patients with a reported history of aspirin intolerance ? --> Please add an algorithm, cause it is missing.

b) Second point: What are current (in the last couple of years) and future upcoming studies including registries, RCT, etc in the field ? This point is missing and is a relevant remark to finish up the review.

Thank you!

Please at the Introduction-Part:

Spell out word CHD -> It is missing at the text.

Author Response

The reviewer’s comments have been greatly appreciated and fully considered in the revision of the manuscript

The submitted review with title: "ASA allergy and desensitization protocols in the management of CAD" describes an important clinical point for our daily life in how to deal with this phenomenon. As a pathology it has been underrated in the last years. However, its relevance is being recently more noted. 

The management of CAD and Aspirin intolerance is challenging, as a proper practical guideline based approach on how to manage it, in this field is still vague. 

The present paragraphs are clearly presented and logically sorted out. 
The authors managed to properly analyse the current literature in this field and bring out this thoroughly thoughted review. The syntax of the manuscript is correctly done.

No major are to be stressed out. However two important points should be addressed to: 

  1. Due to the vast majority of reported adverse reactions to aspirin weighed with the overall benefit of aspirin in CAD patients, we believe aspirin desensitization protocols serve as an excellent method of allowing CAD patients to be more tolerable to aspirin. Therefore by taking the nature of being a review in this fied --> What is your practical appraoch in managing this sensitive matter ? What could be to fullfill the review, a practical management approach to CAD patients with a reported history of aspirin intolerance ? --> Please add an algorithm, cause it is missing.

REPLY 1: Thank you for the comment. We fully agree about the relevance of ASA desensitization for extending the possibility of prescribing ASA to CAD patients and our approach encloses the application of these protocols every time they are feasible. Nevertheless, since they are not validated, so far in guidelines, no algorithm can routinely suggest them as a strategy of choice. This is now better discussed in page 5, lines 201-205.

  1. b) Second point: What are current (in the last couple of years) and future upcoming studies including registries, RCT, etc in the field ? This point is missing and is a relevant remark to finish up the review.

Thank you!

 REPLY 2: Thank you for the comment. We have searched clinicaltrials.gov and currently published protocols of ongoing studies, although detecting no ongoing studies or RCTs on the topic, as now underlined in page 5, line 208.

Reviewer 3 Report

This is well written and properly designed paper. Only one suggestion: title should suggest that presented study is " Review of litarature"

Author Response

This is well written and properly designed paper. Only one suggestion: title should suggest that presented study is " Review of litarature"

We thank the reviewer for the comments, that have greatly acknowledged in the revision of the manuscript. As suggested, in particular, we have updated the title of the manuscript underlying its nature of “review”.